# Global High-Resolution Drought Indices for 1981-2022

Solomon H. Gebrechorkos[1,2], Jian Peng[3,4], Ellen Dyer[1], Diego G. Miralles[5], Sergio M. Vicente-Serrano[6], Chris Funk[7], Hylke E. Beck[8], Dagmawi T. Asfaw[9], Michael B. Singer[10,11], Simon J. Dadson[1,12]

[1]School of Geography and the Environment, University of Oxford, OX1 3QY Oxford, UK
[2]School of Geography and Environmental Science, University of Southampton, Southampton, SO17 1BJ, UK
[3]Department of Remote Sensing, Helmholtz Centre for Environmental Research - UFZ, 04318 Leipzig, Germany
[4]Remote Sensing Centre for Earth System Research – RSC4Earth, Leipzig University, 04103 Leipzig, Germany
[5]Hydro-Climatic Extremes Lab (H-CEL), Ghent University, Ghent, Belgium
[6]Instituto Pirenaico de Ecología, Consejo Superior de Investigaciones Científicas (IPE-CSIC) Zaragoza, Spain
[7]Santa Barbara Climate Hazards Center, University of California, Santa Barbara, USA
[8]Climate and Livability Initiative, King Abdullah University of Science and Technology, Thuwal 23955, Saudi Arabia
[9]School of Geographical Sciences, University of Bristol, Bristol, BS8 1SS, United Kingdom
[10]School of Earth and Environmental Sciences, Cardiff University, Cardiff, CF10 3AT, United Kingdom
[11]Earth Research Institute, University of California, Santa Barbara, USA
[12]UK Centre for Ecology and Hydrology, Wallingford, OX10 8BB UK.

*Correspondence to*: Solomon H. Gebrechorkos (solomon.gebrechorkos@ouce.ox.ac.uk)

**Abstract.** Droughts are among the most complex and devastating natural hazards globally. High-resolution datasets of drought metrics are essential for monitoring and quantifying the severity, duration, frequency and spatial extent of droughts at regional and particularly local scales. However, current global drought indices are available only at a coarser spatial resolution (>50 km). To fill this gap, we developed four high-resolution (5 km) gridded drought records based on the Standardized Precipitation Evaporation Index (SPEI) covering the period 1981-2022. These multi-scale (1-48 months) SPEI indices are computed based on monthly precipitation (P) from the Climate Hazards group InfraRed Precipitation with Station data (CHIRPS, version 2) and Multi-Source Weighted-Ensemble Precipitation (MSWEP, version 2.8) and potential evapotranspiration (PET) from the Global Land Evaporation Amsterdam Model (GLEAM, version 3.7a) and Bristol Potential Evapotranspiration (hPET). We generated four SPEI records based on all possible combinations of P and PET datasets: CHIRPS-GLEAM, CHIRPS-hPET, MSWEP-GLEAM, and MSWEP-hPET. These drought records were evaluated globally and exhibited excellent agreement with observation-based estimates of SPEI, root zone soil moisture, and vegetation health indices. The newly developed high-resolution datasets provide more detailed local information and be used to assess drought severity for particular periods and regions and to determine global, regional, and local trends, thereby supporting the development of site-specific adaptation measures. These datasets are publicly available at the Centre for Environmental Data Analysis (CEDA, https://dx.doi.org/10.5285/ac43da11867243a1bb414e1637802dec) (Gebrechorkos et al., 2023).

## 1. Introduction

Drought is one of the most complex and major natural hazards and it has devastating impacts on the environment, economy, water resources, agriculture and society worldwide (Wilhite et al., 2007; Sternberg, 2011; CRED, 2018; Van Loon, 2015; Sheffield et al., 2012; UNCCD, 2022). The most negative impacts of drought include crop failure, food crisis, famine, malnutrition, and poverty, which lead to loss of life and mass migration (Vicente-Serrano et al., 2012; Haile et al., 2019; Ngcamu and Chari, 2020; Gebrechorkos et al., 2020; UNDRR, 2021). Globally, the occurrence of extreme events such as droughts has increased as a result of the increase in temperature and atmospheric evaporative demand (Sheffield and Wood, 2012; Mukherjee et al., 2018; Van Loon et al., 2022; Wehner et al., 2021). In addition, increased climate variability has increased the frequency and severity of drought events (Sivakumar et al., 2014; Naumann et al., 2018), which has resulted in significant environmental and socioeconomic damage globally (Mukherjee and Mishra, 2021; UNCCD, 2022). Moreover, the occurrence and impact of droughts are aggravated by anthropogenic activities such as land use change and water management and demand (Van Loon et al., 2022; UNCCD, 2019). During the past few decades a number of extreme drought events, with significant impacts, have occurred in different parts of the world such as in 2001-2009 in Southeast Australia (van Dijk et al., 2013; Peng et al., 2019a), 2017 in Europe (García-Herrera et al., 2019), 2015 in East Africa (FEWS-NET, 2015), 2010 in Russia (Spinoni et al., 2015), 2014 in Northern China (Wang and He, 2015), 2012 to 2019 in California (Warter et al., 2021), 2015 to 2017 in South Africa (Baudoin et al., 2017), among others. These trends are expected to continue in the future under the projected change in climate (IPCC, 2014; Naumann et al., 2018; Wehner et al., 2021).

Several indices have been defined to quantify and monitor drought at different spatial and temporal scales. Drought indices such as the Palmer Drought Severity Index (Palmer, 1965), Standardized Precipitation Index (McKee et al., 1993), Standardized Precipitation-Evapotranspiration Index (Vicente-Serrano et al., 2010a), Soil Moisture Deficit Index (Narasimhan and Srinivasan, 2005), Deciles Index (Gibbs, 1967), and Standardized Runoff Index (Shukla and Wood, 2008), among many others, have been developed and remain widely used. A key property of drought indices is their spatial comparability and they must be statistically robust (Keyantash and Dracup, 2002). This is also the main property of the Standardized Precipitation Index (SPI), which was recommended by the WMO (World Meteorological Organization) for identifying and monitoring meteorological droughts in different climates and time periods (Hayes et al., 2011). SPI is computed based on precipitation, which makes it a simple and easy-to-apply for monitoring and prediction of droughts in different parts of the world (e.g., Zhao et al., 2017; Gebrechorkos et al., 2020; Sun et al., 2022; Cammalleri et al., 2022; Vergni et al., 2017; Kumar et al., 2016; Gao et al., 2018; Hao et al., 2014; Lotfirad et al., 2022). However, SPI does not consider other climate variables that might affect drought and are not stationary in the current climate change scenario, particularly those meteorological variables that control the atmospheric evaporative demand (Vicente-Serrano et al., 2020, 2022). In fact, several studies have suggested that global warming has increased drought severity, demonstrated by the increased stress on vegetation and water resources (Lespinas et al., 2010; Liang et al., 2010; Carnicer et al., 2011; Allen et al., 2015; Matiu et al., 2017; Mastrotheodoros et al., 2020). Hence, the Standardized Precipitation Evapotranspiration Index (SPEI; Vicente-Serrano et al., 2010a) accounts also for the role of the

increased atmospheric evaporative demand on drought severity (Vicente-Serrano et al., 2020), being particularly dominant during periods of precipitation deficit.

Similar to SPI, SPEI can be calculated for a range of timescales (1 - 48 months). SPEI calculation requires long-term and high-quality precipitation and atmospheric evaporative demand datasets, which can be obtained from ground stations or gridded data based on reanalysis, satellite and multi-source datasets. High-resolution drought information helps better assess the spatial and temporal changes and variability in drought duration, severity, and magnitude at a much finer scale, which supports the development of site-specific adaptation measures. Globally, the SPEIbase (Vicente-Serrano et al., 2010b) and GPCC Drought Index (Ziese et al., 2014) datasets are available at a relatively coarse spatial resolution; the SPEIbase is available at 0.5° resolution calculated from the Climatic Research Unit (Harris et al., 2020) precipitation and potential evapotranspiration datasets, while the GPCC drought index provided SPEI datasets at a 1.0° spatial resolution for limited timescales (1, 3, 6, 9, 12, 24 and 48 months). These datasets, although useful for long-term assessment, may have limitations for the assessment of drought characteristics at detailed spatial scales.

In this study, four global high-resolution (0.05°) SPEI datasets based on two high-resolution precipitation and two potential evapotranspiration datasets are developed covering the period from 1981 to 2022. The new SPEI datasets are evaluated against coarser spatial resolution SPEI datasets and other variables such as root zone soil moisture and vegetation indices. The precipitation and evapotranspiration datasets used in this study are widely used and generally reliable, although there may be some regions where their reliability is limited. Hence, developing multiple indices using different datasets will help better manage and monitor droughts than using a single dataset (Turco et al., 2020), particularly in data-scarce regions of the world such as in Africa and South America. Using a single dataset can be limiting, as it may not capture the full spectrum of drought characteristics and impacts. In this study, we utilize two distinct precipitation datasets, each with its unique processing methods and data sources. This approach allows for a detailed assessment of results, significantly increasing the reliability and confidence in the drought assessment. Furthermore, these high-resolution global drought indices serve a critical role in assessing drought impacts on a global, regional, and local scale. Such assessments are invaluable for informing the development of site-specific adaptation measures, as they offer a more nuanced understanding of drought events and their consequences on different scales. The manuscript offers a comprehensive overview of the datasets and methodologies employed, outlined in Section 2. Section 3 provides an in-depth analysis of the results, coupled with a detailed discussion. The availability of the newly developed high-resolution datasets, along with relevant links, is covered in Section 4, while Section 5 delivers a concise conclusion.

## 2.  Data and methodology

### 2.1. Datasets

High-resolution precipitation and potential evapotranspiration datasets are selected for developing multi-scale high-resolution SPEI globally (Table 1). To assess the quality of the new high-resolution SPEI (SPEI-HR) we used multiple datasets such as

a coarse resolution global SPEI (SPEI-CR), root-zone soil moisture (SMroot), Vegetation Condition Index (VCI), and Vegetation Health Index (VHI). VCI and VHI are computed based on Vegetation indices such as the Normalized Difference Vegetation Index (NDVI) and temperature datasets (Table 1).

Table 1. Overview of the different datasets used to develop and evaluate the new high-resolution SPEI (SPEI-HR) datasets.

| Short name | Full name and details | Spatial resolution | Temporal coverage | Reference |
|---|---|---|---|---|
| CHIRPS | Climate Hazards group InfraRed Precipitation with Station data version 2.0 (https://data.chc.ucsb.edu/products/CHIRPS-2.0/) | 0.05° | 1981-present | Funk et al. (2015) |
| MSWEP | Multi-Source Weighted-Ensemble Precipitation version 2.8 (https://www.gloh2o.org/mswep/) | 0.1° | 1979-present | Beck et al. (2019) |
| GLEAM PET | Potential evapotranspiration from the Global Land Evaporation Amsterdam Model version 3.7a (https://www.gleam.eu/) | 0.25° | 1980-2022 | Martens et al. (2017); Miralles et al. (2011) |
| GLEAM SMroot | Root-zone soil moisture from the Global Land Evaporation Amsterdam Model version 3.7a (https://www.gleam.eu/) | 0.25° | 1980-2022 | Martens et al. (2017); Miralles et al. (2011) |
| hPET | Hourly potential evapotranspiration from the University of Bristol (https://data.bris.ac.uk/data/dataset/qb8ujazzda0s2aykkv0oq0ctp) | 0.1° | 1981-2022 | Singer et al. (2021) |
| NDVI | Vegetation indices such as the Normalized Difference Vegetation Index from Moderate Resolution Imaging Spectroradiometer (https://ladsweb.modaps.eosdis.nasa.gov/missions-and-measurements/products/MOD13C2) | 0.05° | 2000-present | Didan, Kamel, (2021) |
| Temperature | Land surface temperature data from Moderate Resolution Imaging Spectroradiometer (https://ladsweb.modaps.eosdis.nasa.gov/missions-and-measurements/products/MOD11C3) | 0.05° | 2000-present | Wan, Zhengming et al. (2021) |
| SPEI-CR | Global SPEI database, SPEIbase v2.8, (https://spei.csic.es/database.html) | 0.5° | 1901-2021 | Beguería et al. (2014); Vicente-Serrano et al. (2010a) |

### 2.1.1. Precipitation

The Climate Hazards group InfraRed Precipitation with Station data version 2.0 (CHIRPS) and Multi-Source Weighted-Ensemble Precipitation version 2.8 (MSWEP) precipitation estimates are used to compute SPEI quasi-globally (50S–50N) and globally, respectively.

CHIRPS is a high-resolution quasi-global rainfall product primarily developed for monitoring droughts and global environmental changes (Funk et al., 2015b). CHIRPS provides gauge-satellite precipitation estimates covering most of the globe with a high spatial resolution (0.05°), low bias, and a long period of record. The product is developed by combining satellite-only Climate Hazards group Infrared Precipitation (CHIRP), Climate Hazards group Precipitation climatology (CHPclim, Funk et al., (2015a)), and data from ground stations. CHIRP and CHPclim were developed based on calibrated infrared cold cloud duration (CCD) precipitation estimates and ground station data from the Global Historical Climate Network (GHCN) and other sources. The product is available at the Climate Hazards Center (https://www.chc.ucsb.edu/data/chirps/) on daily, 10-day, and monthly timescales from 1981 to near present. CHIRPS has been evaluated against ground observations and has been widely used in climate, hydrology, water resources studies and for monitoring droughts (e.g., Peng et al., 2020; Pyarali et al., 2022; AL-Falahi et al., 2020; Gebrechorkos et al., 2018, 2019b, a; Mianabadi et al., 2022; Habitou et al., 2020; Ghozat et al., 2022; Gao et al., 2018; Sandeep et al., 2021). The CHIRPS product benefits from homogeneity. The single source of background information (CHIRP), is based on a continuous stream of geostationary satellite thermal infrared observations. These low-bias background fields are then blended with a large set of quality-controlled station observations using a geostatistical blending procedure. The low bias of CHIRP reduces discontinuities associated with changes in station observation networks, which are common in the global south.

MSWEP is a global (all land and oceans) high-resolution (0.1°) precipitation product developed by merging multiple datasets including observed station data (~77,000 stations), reanalyses, and satellite-based rainfall estimates (Beck et al., 2019). The observed station data includes the Global Summary of the Day (GSOD), Global Historical Climatology Network-Daily (GHCN-D), WorldClim, Global Precipitation Climatology Centre (GPCC), and various national databases. The satellite datasets incorporated in MSWEP include Global Satellite Mapping of Precipitation (GSMaP), Climate Prediction Center morphing technique (CMORPH), Tropical Rainfall Measuring Mission (TRMM) Multi-satellite Precipitation Analysis (TMPA-3B42RT), and Gridded Satellite (GridSat). The reanalyses incorporated in MSWEP include the Japanese 55-year Reanalysis (JRA-55) and European Centre for Medium-Range Weather Forecasts (ECMWF) interim reanalysis (ERA-Interim). The data has been evaluated and showed the best correlation compared to other 22 global precipitation datasets (Beck et al., 2017) and widely used in climate, hydrology and for monitoring droughts (e.g., Li et al., 2022; Xu et al., 2019; Li et al., 2023; Guo et al., 2022; Ramsankaran et al., 2018; Gebrechorkos et al., 2022; Swain et al., 2017; Alijanian et al., 2022; Turco et al., 2020). MSWEP is available via the GloH2O website (https://www.gloh2o.org/mswep/) from 1979 - near present on 3-hourly, daily, and monthly time steps.

### 2.1.2. Potential Evapotranspiration

High-resolution potential evapotranspiration (PET) from the Global Land Evaporation Amsterdam Model (GLEAM) and hourly potential evapotranspiration (hPET) are used as input to develop the SPEI datasets. GLEAM is a set of algorithms designed to calculate actual evaporation, PET, evaporative stress, and root-zone soil moisture ( Miralles et al., 2011). The PET from GLEAM v3.7a (GLEAM-PET) is developed based on a Priestley-Taylor equation (Eq. 1) driven by satellite and reanalysis data. The data are available globally from 1980 to 2022 at 0.25° spatial resolution at daily and monthly timescales (https://www.gleam.eu/). The GLEAM data have been extensively evaluated and widely used for global, continental, and regional scale hydro-meteorological applications and drought studies (e.g., Greve et al., 2014; Miralles et al., 2014; Trambauer et al., 2014; Forzieri et al., 2017; Lian et al., 2018; Wartenburger et al., 2018; Zhan et al., 2019; Peng et al., 2019b; Vicente-Serrano et al., 2018).

hPET is a recently developed global high-resolution (0.1°) dataset available from 1981–2022 (Singer et al., 2021). hPET is developed based on ERA5-Land reanalysis and it is openly available from the University of Bristol (https://data.bris.ac.uk/data/dataset/qb8ujazzda0s2aykkv0oq0ctp). Unlike the simple Priestley-Taylor equation used in GLEAM-PET, the FAO Penman-Monteith equation (Eq. 2) is used in hPET (Allan et al., 1998), which, in addition to radiation, temperature and pressure, requires wind speed and dew-point temperature. The hPET data compares closely with the PET calculated based on the Climatic Research Unit (CRU) climate datasets, which are derived from field-based meteorological stations also based on the same FAO Penman-Monteith equation (Singer et al., 2021).

$$PET_{pt} = \alpha * \frac{\Delta*(R_{n-}G)}{\lambda_v*(\Delta+\gamma)} \tag{1}$$

$$\lambda ET_{pm} = \frac{\Delta*(R_{n-}G)+\rho_a*c_p*(\frac{e_s-e_a}{r_a})}{\Delta+\gamma(1+\frac{r_s}{r_a})} \tag{2}$$

Where $\alpha$ is the evaporative coefficient, $\Delta$ is the slope of the saturation vapour pressure-temperature relationship, $R_n$ is the net radiation, $G$ is the soil heat flux, $\lambda$ is the latent heat of vaporization, $\gamma$ is the psychrometric constant, $\rho_a$ is the mean air density at constant pressure, $c_p$ is the specific heat of the air, $(e_s - e_a)$ is the vapour pressure deficit of the air, $r_a$ is aerodynamic resistance, and $r_s$ is the surface resistance.

### 2.1.3. Root Zone Soil Moisture

The GLEAM root zone soil moisture (SMroot) is developed from a multilayer water balance driven by precipitation in which microwave surface soil moisture is assimilated (Martens et al., 2017). The GLEAM SMroot has been validated using observed soil moisture data from more than 2300 soil moisture sensors and 91 eddy-covariance sites (Martens et al., 2017), and inter-compared to other frequently used datasets (Beck et al., 2021).

### 2.1.4. Coarse-resolution Global SPEI datasets

The global coarse resolution (0.5°) SPEI is a widely used dataset for drought analysis (Vicente-Serrano et al., 2010b; Beguería et al., 2014). This coarse-resolution SPEI (SPEI-CR) was developed based on the 0.5° monthly precipitation and PET datasets from the Climate Research Unit (CRU-TS) (Harris et al., 2020). The FAO-56 Penman-Monteith equation is used to compute the CRU-TS PET, which is used to develop the SPEI-CR. In this study, SPEI-CR from 1981-2022 is used to evaluate the SPEI-HR datasets.

### 2.1.5. Vegetation Indices

Vegetation indices such as the Normalized Difference Vegetation Index (NDVI), Vegetation Condition Index (VCI), and Vegetation Health Index (VHI) are used to investigate drought conditions (Peng et al., 2020; Pyarali et al., 2022). Compared to NDVI and VCI, VHI includes the effect of temperature on vegetation health. In this study, high-resolution (0.05°) NDVI (MOD13C2) and Land Surface Temperature (MOD11C3) data from Moderate Resolution Imaging Spectroradiometer (MODIS) for the period 2000-2022 are used. VHI is computed based on NDVI and Land Surface Temperature (LST) for the period 2000-2022 following similar work done for Europe (Bachmair et al., 2018). To compute VHI (Eq. 5), first, the Vegetation Condition Index (VCI, Eq. 3) and Temperature Condition Index (TCI, Eq. 4) are computed using the methods developed by Kogan (1995).

$$\text{VCI} = \frac{\text{NDVI} - NDVI_{min}}{\text{NDVI}_{max} - NDVI_{min}} * 100 \tag{3}$$

$$\text{TCI} = \frac{\text{T}_{max} - T}{T_{max} - T_{min}} * 100 \tag{4}$$

$$\text{VHI} = \alpha * \text{VCI} + (1 - \alpha) * \text{TCI} \tag{5}$$

where $NDVI$ is the monthly NDVI and $NDVI_{min}$ and $NDVI_{max}$ are the minimum and maximum NDVI values in the time series, respectively. In addition, $T$ is the monthly average temperature, and $T_{min}\ and\ T_{max}$ are the minimum and maximum values of average temperature, respectively. Finally, the $\alpha$ is used as a constant value (0.5) in the computation of VHI.

### 2.2. Methods

The Standardized Precipitation-Evapotranspiration Index (SPEI) (Vicente-Serrano et al., 2010a; Beguería et al., 2014) is a multiscalar drought index which combines the effect of atmospheric evaporative demand and precipitation to quantify the intensity and magnitude droughts over a given period and on different spatial scales (station to global scales). SPEI has been used in a wide range of applications in hydrology, climate and agriculture (e.g., Pyarali et al., 2022; Peng et al., 2020; Wang et al., 2021; Gebrechorkos et al., 2020; Naumann et al., 2018; Mohammed et al., 2022). The steps in computing SPEI include the development of high-quality PET data and the calculation of the difference between supply and atmospheric water demand (Precipitation – PET) at different time scales (1–48 months). These differences are transformed into a normal standard

distribution using a log-logistic probability distribution fit in order to have comparable values among periods and regions. The choice of the log-logistic distribution for SPEI is based on previous research (Vicente-Serrano et al., 2010a; Beguería et al., 2014), which demonstrated its superior ability to generate SPEI series with standardized properties (mean = 0, SD = 1) compared to other probability distributions. The log-logistic distribution involves three key parameters: α (scale), β (shape), and γ (origin), which are estimated using the robust and straightforward L-moment procedure. Further details on the parameter computation process can be found in Vicente-Serrano et al. (2010). The wet and dry categories according to the SPEI are summarised in Table 2. In this study, we computed SPEI datasets for the period 1981-2022 using two sets of precipitation and potential evapotranspiration (PET) data. The precipitation datasets CHIRPS and MSWEP were paired with the PET datasets GLEAM PET and hPET. This resulted in the following SPEI indices: MSWEP and GLEAM PET (MSWEP_GLEAM), MSWEP and hPET (MSWEP_hPET), CHIRPS and GLEAM PET (CHIRPS_GLEAM), and CHIRPS and hPET (CHIRPS_hPET). A flow chart is provided in the supplementary material (Figure S1). MSWEP, GLEAM and hPET are spatially interpolated to a 0.05°resolution using a bilinear interpolation to match with CHIRPS. Finally, four SPEI datasets are developed based on CHIRPS and GLEAM (CHIRPS_GLEAM), CHIRPS and hPET (CHIRPS_hPET), MSWEP and GLEAM (MSWEP_GLEAM), and MSWEP and hPET (MSWEP_hPET).

**Table 2:** SPEI values and their wet and dry categories

| SPEI categories | SPEI values (Danandeh Mehr et al., 2020) | SPEI values (Agnew, 2000) |
|---|---|---|
| Extremely wet | >1.83 | |
| Very wet | 1.43 to 1.82 | |
| Moderate wet | 1.0 to 1.42 | |
| Near Normal | -0.99 to 0.99 | |
| Moderately dry | -1.0 to -1.42 | -0.84 to -1.27 |
| Severely dry | -1.43 to -1.82 | -1.28 to -1.64 |
| Extremely dry | < -1.83 | < -1.65 |

The developed high-resolution global SPEI datasets (SPEI-HR) based on CHIRPS_GLEAM, CHIRPS_hPET, MSWEP_GLEAM, and MSWEP_hPET are evaluated against SPEI-CR, SMroot, VCI, and VHI at a global and quasi-global scales. For comparison, the SPEI-HR is aggregated to the resolution of SPEI-CR (0.5°). In addition, the SMroot is bilinearly interpolated to match the resolution of SPEI-HR. For direct comparison with SPEI, the SMroot, VCI, and VHI are normalised by subtracting the long-term mean and dividing by the standard deviation. Before normalising the VCI and VHI values, the seasonal cycle is removed from the time series. Standardising absolute values allows better comparison of different datasets and has been applied in various studies to compare different drought indices (Pyarali et al., 2022; Peng et al., 2020; Anderson et al., 2011; Mu et al., 2013; Zhao et al., 2017). The comparison between the SPEI-HR and VHI is performed for the period

2000-2022. The agreement between the SPEI-HR and SPEI-CR, SMroot, VCI, and VHI is assessed using Pearson's correlation coefficient.

### 3. Results and Discussion

#### 3.1. Agreement between SPEI-HR datasets and SPEI-CR

SPEI-HR, compared to SPEI-CR, provides more local information and spatial detail on drought conditions (Figure 1). Figure 1 shows an example of severe droughts in South Africa in 2015 and Australia in 2019. In December-February 2015-2016, South Africa faced a severe drought driven by one of the strongest El Niño events in the last 5 decades, which caused severe consequences for food security in the region (Funk et al., 2018, 2016). In addition, November 2019 was the driest month in Australia with the lowest rainfall record (Funk, 2021). According to the Australian Government Bureau of Meteorology

(Australian Government Bureau of Meteorology, 2023), November 2019 was the driest month across most of the country with lower rainfall and drier soil moisture records. The severe drought events in South Africa and Australia are very well reproduced by the SPEI-CR (Figure 1). The SPEI-HR, compared to the SPEI-CR, clearly depicted the wet and drier part of Australia in November 2019. In Australia, for example, the wet events in the northern part of Western Australia and the central part of New South Wales states are well represented by the SPEI-CR. Similarly, the drought in South Africa in 2015 is well

represented by the SPEI-HR as reported by Funk et al. (2018). However, wet events (SPEI up to 2.4) are shown by the SPEI-CR in the northern part of Mozambique. Overall, high-resolution information is more useful for managing and monitoring droughts at a local scale than SPEI-CR, which exhibits much smoother patterns due to the interpolation of station observations (McRoberts and Nielsen-Gammon, 2012; Santini et al., 2023; Park et al., 2017; Jung et al., 2020).

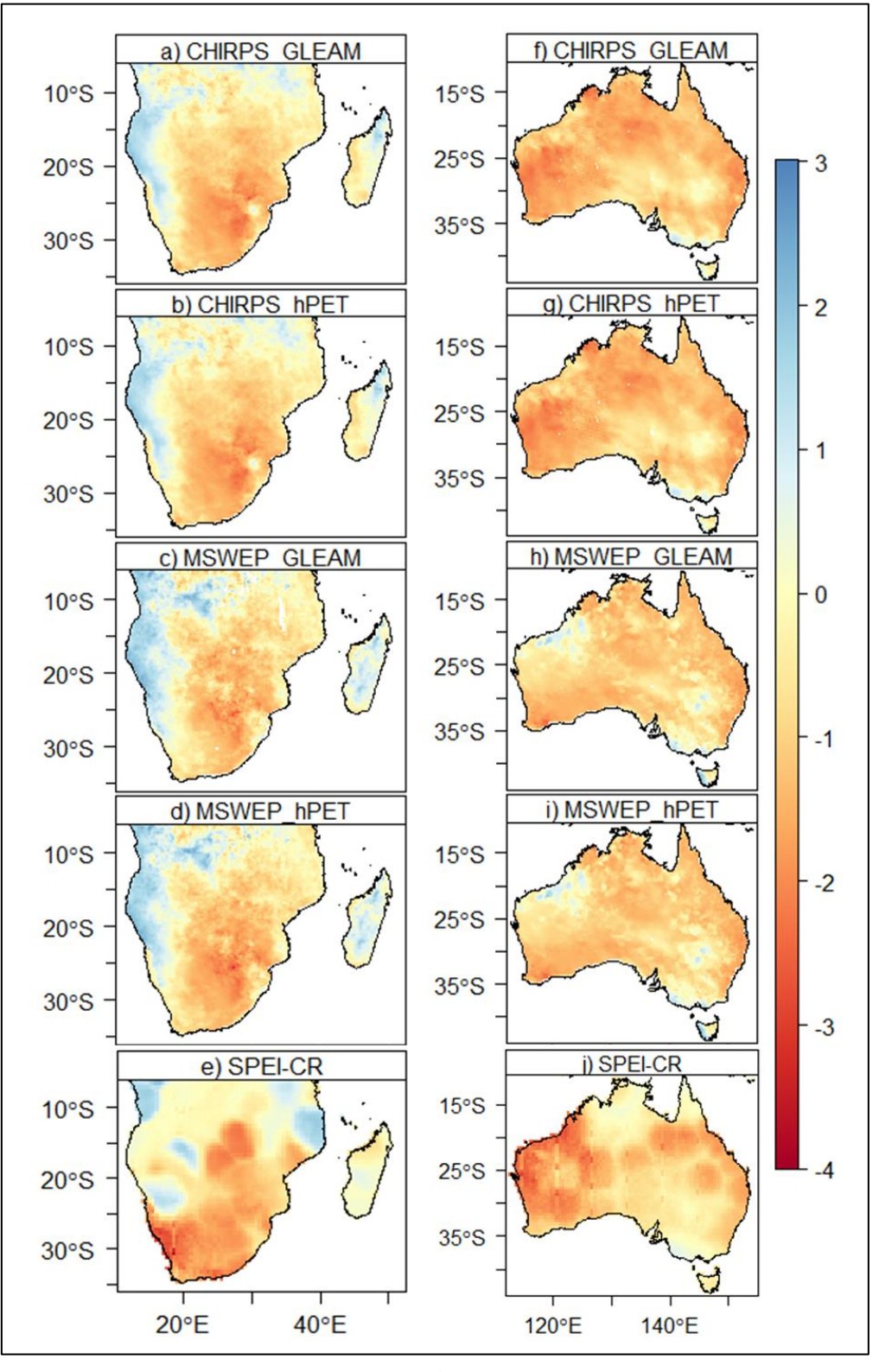

**Figure 1: An example of 1-month SPEI for December 2015 in South Africa and November 2019 in Australia based on CHIRPS_GLEAM (a and f) CHIRPS_hPET (b and g), MSWEP_GLEAM (c and h), MSWEP_hPET (d and i) and SPEI-CR (e and j). The spatial resolution of SPEI-CR is 0.5°.**

Further, the SPEI-HR datasets are temporally evaluated using SPEI-CR during the period 1981-2022. For 01-month SPEI, for example, the correlation between CHIRPS_GLEAM, CHIRPS_hPET, MSWEP_GLEAM, and MSWEP_hPET and SPEI-CR
is significant (R> 0.5, p < 0.1) in large parts of the world (Figure 2). The correlation between the SPEI-HR and SPEI-CR in the USA, Europe, Asia, and Australia is > 0.8. However, it is lower in the tropical and arid regions of Africa and South America. The lower correlations reflect a lower data quality in these regions due to a lack of station observations (Menne et al., 2012) and the frequent occurrence of difficult-to-predict (using models) and measure (using gauges) intense, localized convective storms (Feng et al., 2021).

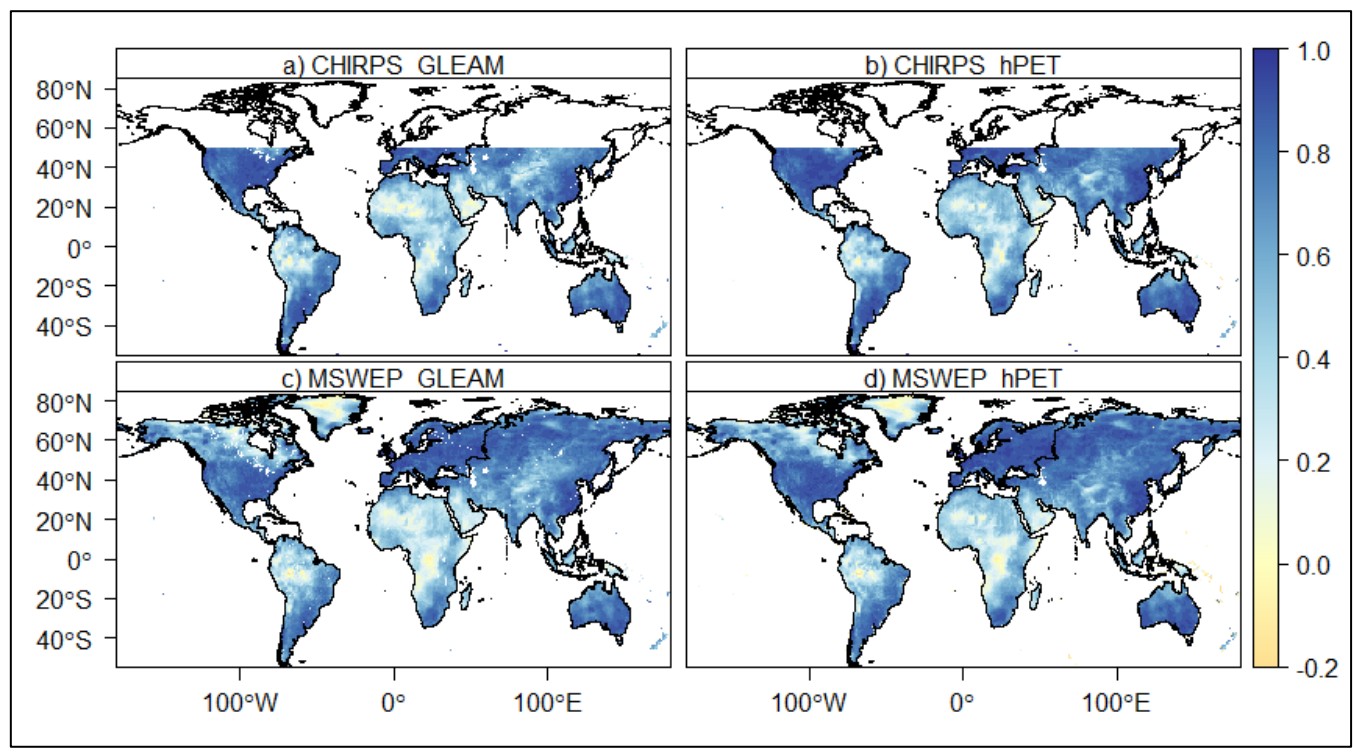

**Figure 2: Temporal correlation between the new high-resolution SPEI (SPEI-HR) based on (a) CHIRPS_GLEAM, b) CHIRPS_hPET, c) MSWEP-GLEAM, and d) MSWEP-hPET and the coarse resolution SPEI (SPEI-CR) for SPEI-01 month.**

The 12-month SPEI from SPEI-HR and SPEI-CR also show a higher correlation in large parts of the world (Figure 3). The correlation between SPEI-HR and SPEI-CR is very high in USA, Asia, and Australia, but lower in central Africa and north
part of South America. In Africa and South America, CHIRPS-GLEAM and CHIRPS-hPET show a higher correlation with SPEI-CR compared to MSWEP-GLEAM and MSWEP-hPET. This higher correlation for CHIRPS may be due to the larger number of stations incorporated in CHIRPS than in MSWEP. The positive correlation increases with an increase in SPEI time scales. This indicates the cancellation of random errors, as the integration period increases. However, there is a low correlation

between SPEI-HR and SPEI-CR in parts of Central Africa and Northern South America. The lower correlation in the tropics (e.g., South America and Africa) is likely due to the very limited number of ground observations used in the development of the CRU-TS precipitation (Harris et al., 2020), which is used to develop the SPEI-CR. In line with this study, a lower correlation between the global drought probabilistic index developed from several global precipitation datasets and the MSWEP-based SPI index was reported in South America and Africa (Turco et al., 2020). These same areas have been shown to be areas of substantial disagreement for wet season precipitation totals (Funk et al., 2019b). In tropical regions with heavy convective precipitation, satellite inputs can produce very different results than station-only precipitation estimations.

The lower correlation in arid areas can also be due to the uncertainty of the forcing datasets, very few station observations, very few events, and intense, localized convective systems, which are difficult to measure and predict in these areas.

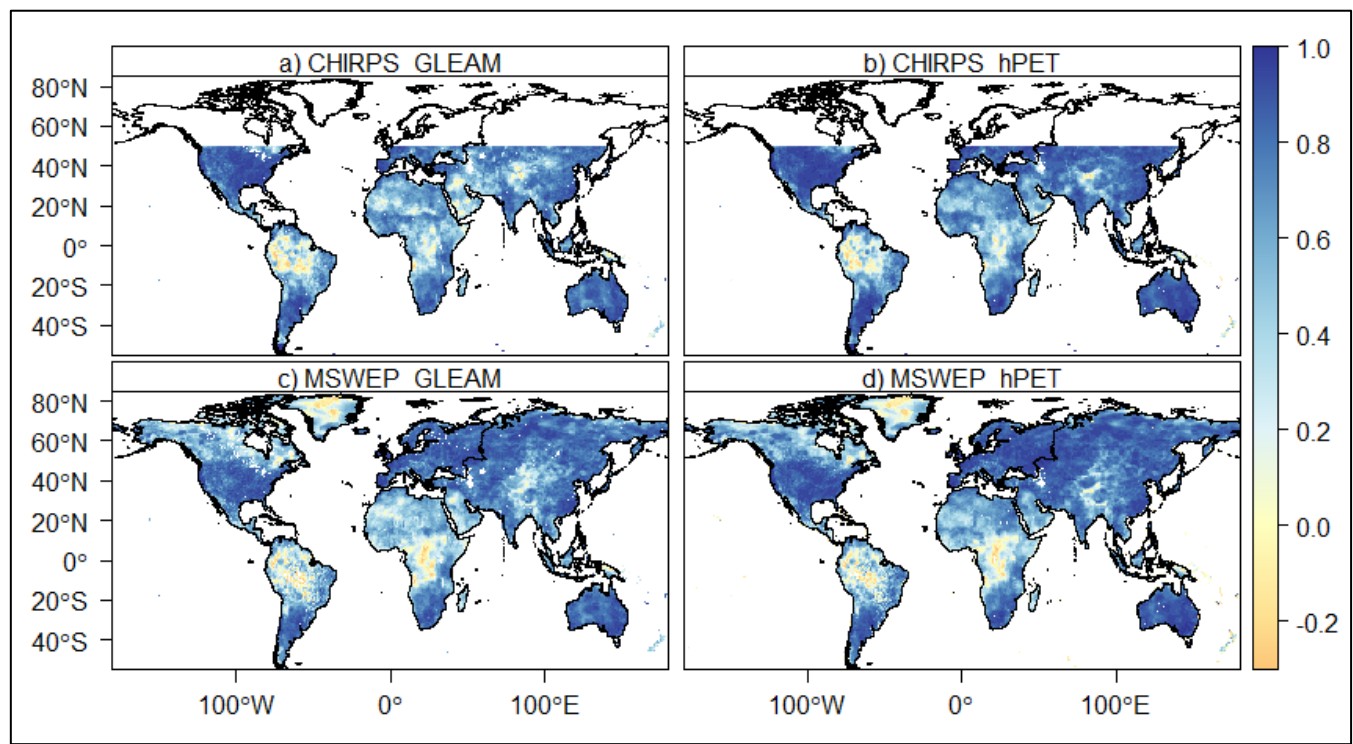

**Figure 3: Temporal correlation between the new high resolutions SPEI (SPEI-HR) based on (a) CHIRPS_GLEAM, b) CHIRPS_hPET, c) MSWEP-GLEAM, and d) MSWEP-hPET and the coarse resolution SPEI (SPEI-CR) for SPEI-12 month.**

### 3.2. Agreement between SPEI-HR and regional SPEI datasets

The new dataset, SPEI-HR, is compared to other high-resolution SPEI datasets in Central Asia (Pyarali et al., 2022) and Africa (Peng et al., 2020). The Central Asia SPEI (CA-SPEI) was developed based on CHIRPS and an older version of GLEAM-PET for the period from 1981-2018. CA-SPEI exhibits a stronger correlation with CHIRPS_GLEAM than with CHIRPS_hPET, MSWEP_hPET, and MSWEP_GLEAM (Figure S2). The correlation between CA-SPEI and

CHIRPS_GLEAM exceeds 0.5. The agreement between CA-SPEI and CHIRPS_hPET, MSWEP_hPET, and MSWEP_GLEAM is high in Kazakhstan but lower in Kyrgyzstan. This variation may be attributed to the differences in the performance of MSWEP and CHIRPS, in addition to variations in PET. According to Peña-Guerrero et al. (2022) MSWEP effectively represents the temporal dynamics of precipitation, while CHIRPS captures the distribution and volume of precipitation in Central Asia. Similarly, SPEI-HR demonstrates a correlation with the Pan African high-resolution SPEI (AF-SPEI) dataset (Peng et al., 2020). In Africa, both CHIRPS_GLEAM and CHIRPS_hPET exhibit a higher correlation (> 0.75) with AF-SPEI compared to MSWEP_GLEAM and MSWEP_hPET for the period 1981-2016 (Figure S3). AF-SPEI, similar to CA-SPEI, was developed based on CHIRPS and an older version of GLEAM PET (version 3a). On the whole, the correlation between SPEI-HR and AF-SPEI exceeds 0.45 in Africa.

### 3.3. Agreement between SPEI-HR and root zone soil moisture

SPEI-HR is also evaluated using root zone soil moisture (SMroot) data obtained from GLEAM. Figure 4, for example, shows the temporal correlation between the 6-month SPEI computed based on the CHIRPS-GLEAM, CHIRPS-hPET, MSWEP-GLEAM and MSWEP-hPET and SMroot. The SPEI-HR shows a reasonably good agreement with the SMroot with a correlation greater than 0.5 in large parts of the world except in some parts of the tropical and subtropical regions. Compared to other parts of the world, the correlation between SPEI-HR and SMroot is lower in the tropical part of Africa. The global average SMroot time series also shows a good agreement with SPEI-HR compared to SPEI-CR. According to Peng et al. (2020), high-resolution SPEI datasets agree well with SMroot at 3-month and 6-month timescales. The SPEI-CR, particularly after 2000, shows a more negative SPEI compared to the SMroot and SPEI-HR (Figure 4e). The difference in SPEI-CR (Figure 4e) compared to SMroot and SPEI-HR can be due to the reduction in the number of stations in the CRU-TS dataset (Funk et al., 2019a). The regional average time series of SPEI-06 and SMroot also show the deviation of the SPEI-HR and SPEI-CR, particularly in Africa, South America and Asia (Figure 5). In Europe and the USA, the regional average 6-month SPEI and SMroot show a similar pattern of wet and dry events. However, in Africa (South America), the SPEI-CR before (after) 2000 shows much drier events than SMroot and SPEI-HR. Similarly, MSWEP-hPET also shows more extreme wet events than SMroot in Africa before 2000.

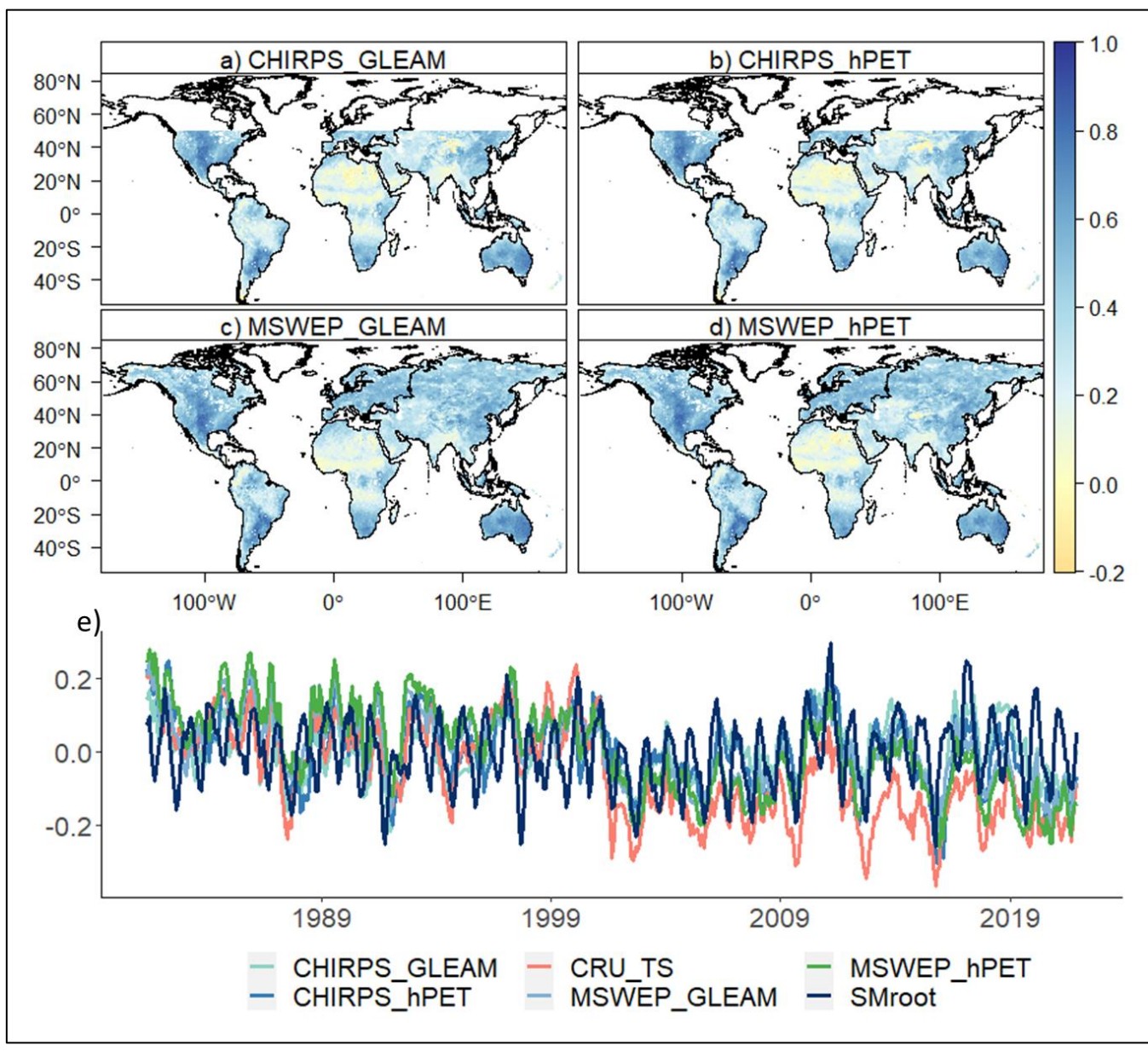

**Figure 4: Temporal correlation between SMroot and 6-month SPEI based on a) CHIRPS_GLEAM, b) CHIRPS_hPET, c) MSWEP_GLEAM, and d) MSWEP_hPET. The lower panel (e) shows a global average (only up to 50°N) time series of SMroot and 6-month SPEI based on CHIRPS_GLEAM, CHIRPS_hPET, MSWEP_GLEAM, MSWEP_hPET and SPEI-CR.**

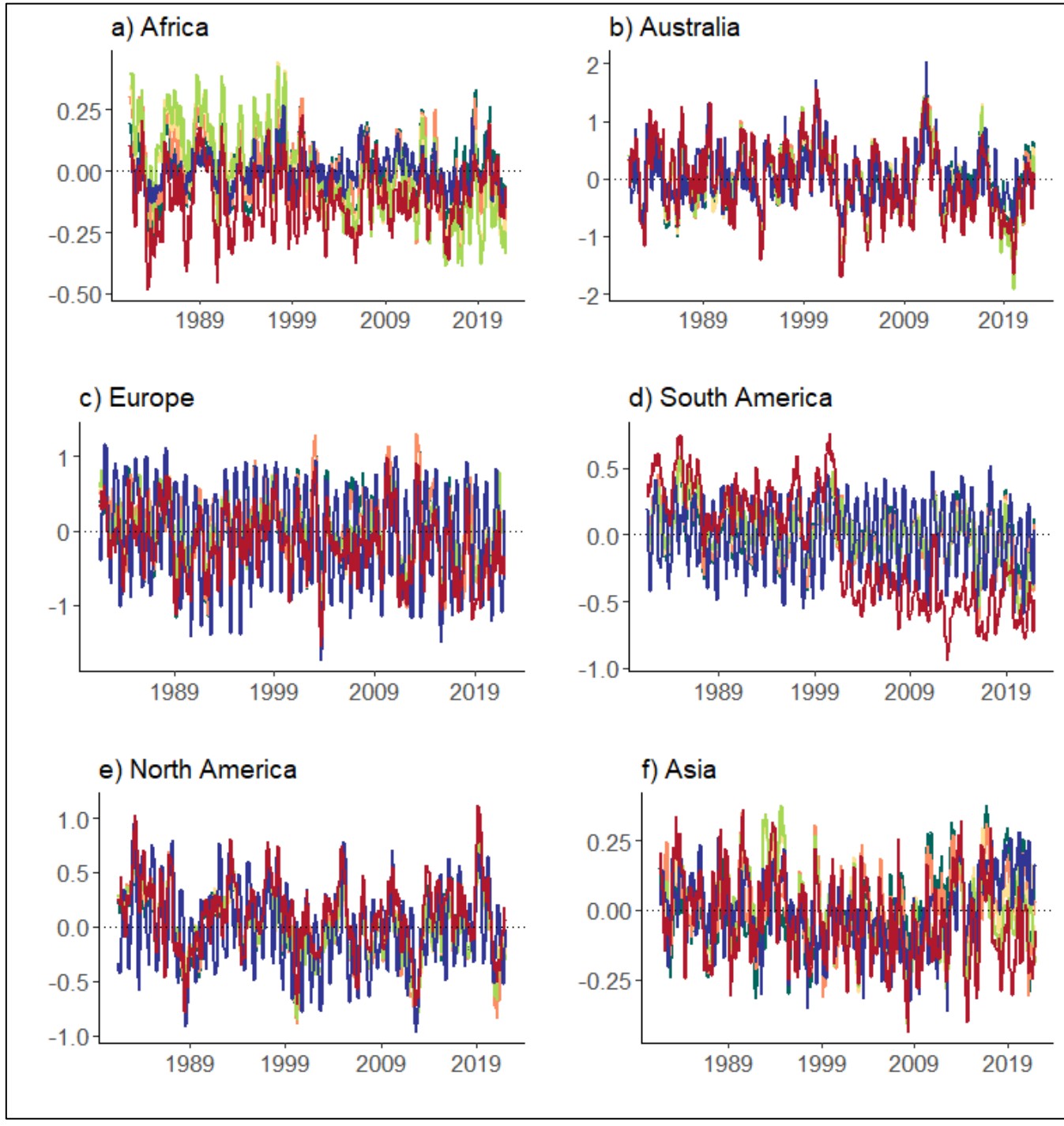

Figure 5: Regional average time series of SMroot and 6-month SPEI based on CHIRPS_GLEAM, CHIRPS_hPET, MSWEP_GLEAM, MSWEP_hPET, and SPEI-CR averaged over a) Africa, b) Australia, c) Europe, d) South America, e) North America (for USA only), and f) Asia.

### 3.4. Agreement between SPEI-HR and vegetation indices

Vegetation indices are used to determine patterns of agricultural drought and they are popular for drought monitoring (Zuhro et al., 2020; Bento et al., 2018; Zeng et al., 2022). Remotely sensed vegetation indices provide additional information for monitoring and early warning of agricultural droughts (Bachmair et al., 2018). Figure 6 shows the correlation between VHI and the 6-month SPEI based on CHIRPS_GLEAM, CHIRPS_hPET, MSWEP_GLEAM, and MSWEP_hPET. The agreement between VHI and MSWEP_GLEAM and MSWEP_hPET is lower in the polar and subpolar regions of North America and Asia. However, there is a positive relationship in other parts of the world such as in South America, Africa, Australia and the east and southern parts of Asia. In addition, the VHI showed a better correlation with SPEI-HR compared to VCI. As described in Section 2.1, VCI is based on NDVI and does not consider explicitly the effect of temperature on vegetation health; yet, VCI shows a positive correlation in different parts of the world, particularly in tropical and subtropical zones (Figure 7).

Results agree with previous studies, which showed a good relationship between vegetation indices and SPEI (Vicente-Serrano et al., 2018; Törnros and Menzel, 2014; Pyarali et al., 2022; Peng et al., 2020). Lower correlations between SPEI-HR and VHI and VCI can be due to the complex nature of vegetation physiological processes and the effect of other climate and environmental drivers (Nemani et al., 2003). The time lag between precipitation and NDVI might also affect the lower correlation as compared to SMroot (Funk and Brown, 2006; Seddon et al., 2016; Papagiannopoulou et al., 2017; Wu et al., 2015). Previous studies have used NDVI to evaluate SPEI (Rojas et al., 2011; Vicente-Serrano et al., 2018; Törnros and Menzel, 2014; Pyarali et al., 2022; Peng et al., 2020). However, compared to VCI, VHI gives a better correlation with SPEI, which might be due to the consideration of the effect of warming on drought and vegetation health.

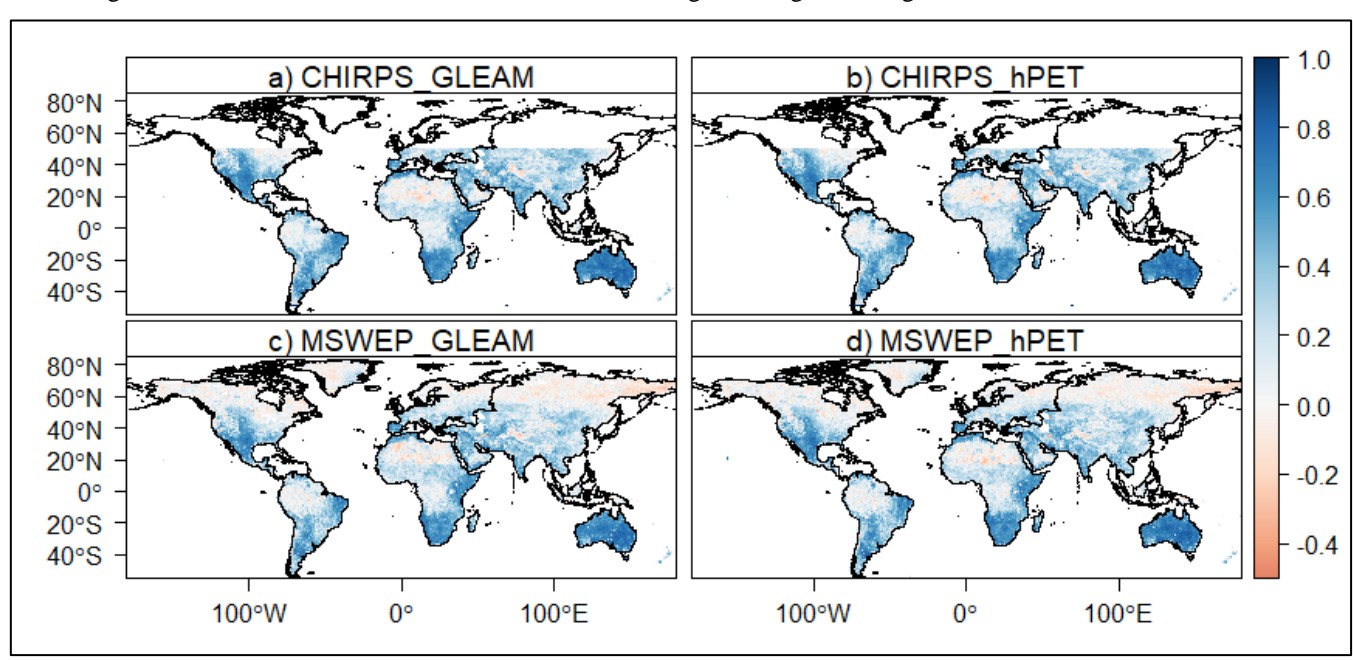

**Figure 6: Temporal correlation between Vegetation Health Index (VHI) and SPEI-6 month based on a) CHIRPS_GLEAM, b) CHIRPS_hPET, c) MSWEP_GLEAM, and d) MSWEP_hPET.**

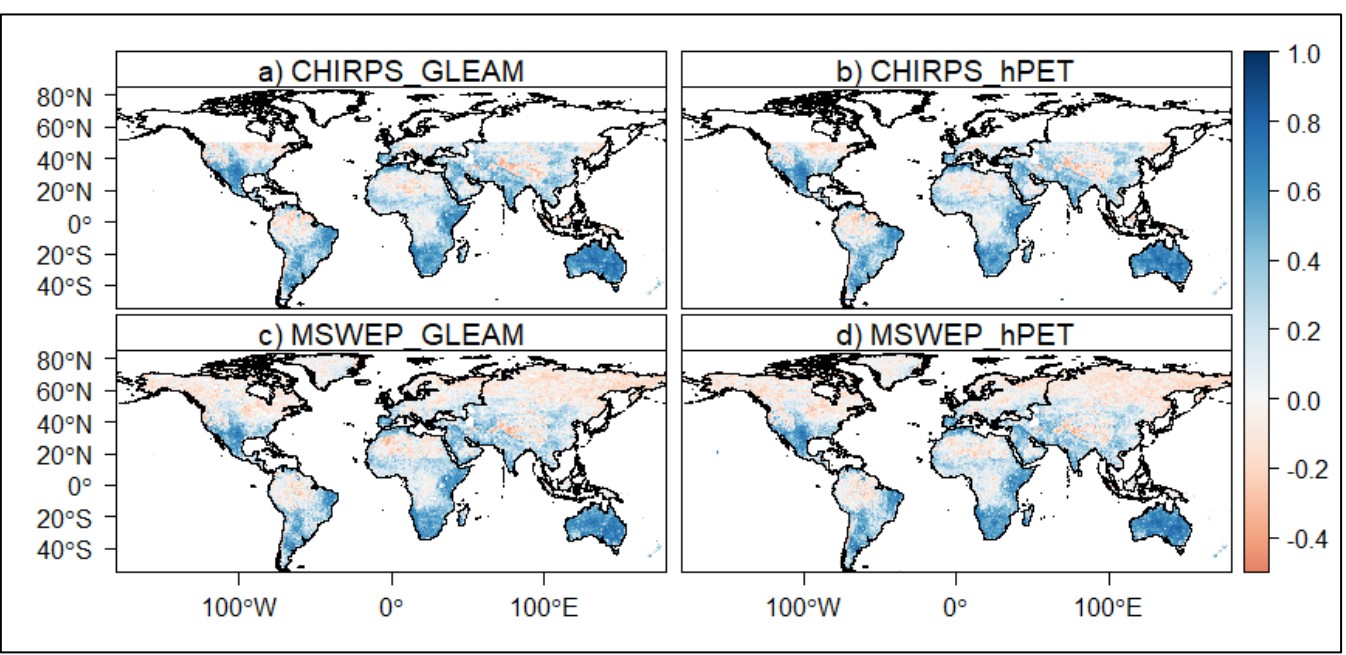

**Figure 7: Temporal correlation between Vegetation Condition Index (VCI) and SPEI-6 months based on a) CHIRPS_GLEAM, b) CHIRPS_hPET, c) MSWEP_GLEAM, and d) MSWEP_hPET.**

### 3.5. Uncertainties in SPEI-HR

This high-resolution SPEI dataset will serve as a basis for drought analysis and the development of effective mitigation and adaptation strategies at a local scale. However, it is important to consider the uncertainties in the datasets due to the uncertainties in the precipitation and PET input datasets. The uncertainty in precipitation and PET, which has its inherent uncertainties, stemming from measurement errors, interpolation methods, and data gaps, can propagate and amplify these uncertainties in SPEI datasets. To reduce the uncertainties, the precipitation datasets employed in this study are widely recognized and have been extensively used in hydroclimatology research, including the development of drought indices (Pyarali et al., 2022; Peng et al., 2020; Hendrawan et al., 2022; Ma et al., 2023). Furthermore, a global-scale evaluation of 22 precipitation datasets has demonstrated the reliability of MSWEP and CHIRPS (Beck et al., 2017). Additionally, the GLEAM model output, particularly evapotranspiration, exhibits lower absolute and relative errors compared to other datasets such as Global Land Data Assimilation System (GLDAS) and MODIS Global Evapotranspiration Project (MOD16) (Khan et al., 2018). According to Laimighofer and Laaha (2022) the observation period used to estimate SPEI and SPI distribution parameters introduces uncertainties. These uncertainties tend to significantly reduce when shifting from 20 to 60 years of records, as the parameters stabilize. For instance, increasing the record length from 20 to 60 years reduces the total variance by 58%, although further improvements are not observed beyond 60 years. Additionally, the selection of the distribution also

significantly reduces uncertainty by 23% (Laimighofer and Laaha, 2022). For SPEI, the log-logistic distribution demonstrated its superior ability to generate SPEI series compared to other probability distributions (Vicente-Serrano et al., 2010a; Beguería et al., 2014), which lowers the uncertainties. In addition to the length of observation and distribution method, the parameter estimation method can lead to uncertainties. The estimation of parameters depends on the choice of the distribution method but is often minimal when compared to the uncertainties from input precipitation and PET (Tallaksen and Lanen, 2004; Laimighofer and Laaha, 2022). Overall, it is well-stated that the source of uncertainties in drought indices can be notably reduced by using longer periods of high-quality precipitation and PET datasets and by selecting appropriate distribution methods.

## 4. Data Availability

The new high-resolution (0.05°) global drought indices (SPEI-HR) based on CHIRPS_GLEAM, CHIRPS_hPET, MSWEP_GLEAM, and MSWEP_hPET are freely available at the Centre for Environmental Data Analysis (CEDA; https://dx.doi.org/10.5285/ac43da11867243a1bb414e1637802dec) and on JASMIN (/badc/hydro-jules/data/Global_drought_indices) (Gebrechorkos et al. 2023). JASMIN is a unique data-intensive HPC system for environmental science (https://jasmin.ac.uk/). For getting access to JASMIN please follow this link https://help.jasmin.ac.uk/article/189-get-started-with-jasmin. The data is available under four directories (CHIRPS_GLEAM, CHIRPS_hPET, MSWEP_GLEAM, and MSWEP_hPET) containing spei01 to spei48 months for the period 1981-2022. The SPEI data covers (land only) all longitudes and from -50° to 50° latitude for CHIRPS_GLEAM and CHIRPS_hPET and from -55° to 85° for MSWEP_GLEAM and MSWEP_hPET. The size of each SPEI file (i.e., NetCDF) is between 5 to 9 gigabytes.

## 5. Conclusion

We produced four global high-resolution (0.05°) and long-term (1981-2022) drought datasets using the SPEI. SPEI is a multi-scale drought index used for drought monitoring and to assess the duration and severity of droughts at different scales (global to local scale). Here, SPEI is computed based on high-resolution precipitation from CHIRPS and MSWEP and potential evapotranspiration (PET) from GLEAM and hPET. The GLEAM and hPET are developed based on Priestley-Taylor and FAO Penman-Monteith equations, respectively. These new high-resolution global SPEI datasets (SPEI-HR) are validated using observation-based coarse-resolution SPEI datasets (SPEI-CR), regional scale high-resolution SPEI datasets, root zone soil moisture (SMroot), Vegetation Condition Index (VCI), and Vegetation Health Index (VHI). The good agreement between the SPEI-HR and SPEI-CR, SMroot, VCI, and VHI confirms the suitability of the data for assessing and monitoring droughts. In addition, SPEI-HR provides more local information as it is available at a much higher spatial resolution (0.05°) compared to currently available global drought indices such as SPEI-CR. Overall, these new multi-time scale (1 - 48 months) SPEI datasets

can be used for drought monitoring and assessing their severity and duration at a local and global scale allowing the development of site-specific adaptation measures.

**Author contributions**

SD led the project and conceived the study with input from all authors. SG developed the drought datasets and drafted the

380 manuscript. CF, HB, and DA and MS, developed CHIRPS, MSWEP and hPET datasets, respectively. SD, JP, DM, and SV supported the generation of the dataset and the analysis of the results. All authors contributed to the development of the manuscript.

**Competing interests**

The authors declare that they have no conflict of interest.

**Acknowledgements**

We would like to thank the UK Foreign, Commonwealth and Development Office (FCDO) and UK Natural Environment Research Council (NERC) for providing financial support. We also thank the Centre for Environmental Data Analysis (CEDA) for storing the SPEI-HR datasets.

**Financial support**

The work is supported by the UK Foreign, Commonwealth and Development Office (FCDO) for the benefit of developing countries (Programme Code 201880) and the UK's Natural Environment Research Council (NERC; NE/S017380/1).

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
