# Peer review of "Global High-Resolution Drought Indices for 1981-2022"

_Earth System Science Data, 2023_

## Author Comment (AC3)

Reply to Referee #1

We appreciate the reviewer for dedicating their time and providing valuable, detailed feedback. We have diligently addressed all of the comments and discussed them as follows.

This manuscript released the global 5km drought indices for 1981-2022. This is a very meaningful work because it is a high-resolution dataset of drought. However, some other problems in the manuscript are still concerned in the following:

Authors´ response:

- Thank you for recognizing our efforts and providing feedback.

1. In Figure 5, I suggest the authors to replace (e) with North America.

Authors´ response:

- Thank you. Figure 5e is updated as recommended.

2. A flow chart of generating this dataset could help the future readers.

Authors´ response:

- Thank you for the recommendation. We have included a textual explanation of the workflow in the methodology section and have also added a flowchart in the supplementary materials to balance the number of figures

3. More details on the method should be exposed.

Authors´ response:

- Thank you for your valuable recommendation. We have enhanced the methodology section by incorporating a flowchart and providing a detailed explanation of the steps used in developing the datasets, including the selection of log-logistic probability distribution.

4. One advantage of this dataset is high resolution. Therefore, the examples of spatial details could be shown and compared with other datasets.

Authors´ response:

- We also considered this a valuable suggestion, thanks to the reviewer's recommendation. As a result, we have incorporated two high-resolution datasets into the evaluation for Central Asia and Africa (see the attached figures). Both the African and Central Asian datasets have been derived from CHIRPS data and exhibit a very high correlation when compared to MSWEP.

[Figure]

Figure. Correlation between the new high resolutions SPEI (SPEI-HR) based on (a) CHIRPS_GLEAM, b) CHIRPS_hPET, c) MSWEP-GLEAM, and d) MSWEP-hPET and Central Asia's high resolution SPEI dataset (Pyarali et al., 2022) for SPEI-1month for the period 1981–2018.

[Figure]

Figure. Correlation between the new high resolutions SPEI (SPEI-HR) based on (a) CHIRPS_GLEAM, b) CHIRPS_hPET, c) MSWEP-GLEAM, and d) MSWEP-hPET and the Pan African high resolution (0.05°) SPEI dataset (Peng et al., 2020) for SPEI-1month for the period 1981–2016.

5. The organization of this manuscript should be added to the end of the introduction.

Authors´ response:

- Thank you. Based on your recommendation we have added the organization of the manuscript at the end of the introduction as: "The manuscript provides a detailed description of the datasets used and the methods in Section 2. Section 3 presents a thorough analysis of the results and offers a discussion of these results. The availability, including links to the newly developed high-resolution datasets, is described in Section 4, while Section 5 provides a clear conclusion of the work."

**Reference**

Peng, J., Dadson, S., Hirpa, F., Dyer, E., Lees, T., Miralles, D. G., Vicente-Serrano, S. M., and Funk, C.: A pan-African high-resolution drought index dataset, Earth System Science Data, 12, 753–769, https://doi.org/10.5194/essd-12-753-2020, 2020.

Pyarali, K., Peng, J., Disse, M., and Tuo, Y.: Development and application of high resolution SPEI drought dataset for Central Asia, Sci Data, 9, 172, https://doi.org/10.1038/s41597-022-01279-5, 2022.

---

## Author Response (AR1)

**Reply to Referee #1**

We appreciate the reviewer for dedicating their time and providing valuable, detailed feedback. We have diligently addressed all of the comments and discussed them as follows.

This manuscript released the global 5km drought indices for 1981-2022. This is a very meaningful work because it is a high-resolution dataset of drought. However, some other problems in the manuscript are still concerned in the following:

Authors´ response:

- Thank you for recognizing our efforts and providing feedback.

1. In Figure 5, I suggest the authors to replace (e) with North America.

Authors´ response:

- Thank you. Figure 5e is updated as recommended.

2. A flow chart of generating this dataset could help the future readers.

Authors´ response:

- Thank you for the recommendation. We have included a textual explanation of the workflow in the methodology section and have also added a flowchart in the supplementary materials to balance the number of figures.

**Author's changes:**

- **we have added the following text to the methodology and provided a flowchart in the supplementary material**
  *"In this study, we computed SPEI datasets for the period 1981-2022 using two sets of precipitation and potential evapotranspiration (PET) data. The precipitation datasets CHIRPS and MSWEP were paired with the PET datasets GLEAM PET and hPET. This resulted in the following SPEI indices: MSWEP and GLEAM PET (MSWEP_GLEAM), MSWEP and hPET (MSWEP_hPET), CHIRPS and GLEAM PET (CHIRPS_GLEAM), and CHIRPS and hPET (CHIRPS_hPET). A flow chart is provided in the supplementary material (Figure S1)."*

3. More details on the method should be exposed.

Authors´ response:

- Thank you for your valuable recommendation. We have enhanced the methodology section by incorporating a flowchart and providing a detailed explanation of the steps used in developing the datasets, including the selection of log-logistic probability distribution. For example, we have explained why the log-logistic distribution is used: as "(**author's changes)**
- *The choice of the log-logistic distribution for SPEI is based on previous research (Vicente-Serrano et al., 2010a; Beguería et al., 2014), which demonstrated its superior ability to generate SPEI series with standardized properties (mean = 0, SD = 1) compared to other probability distributions. The log-logistic distribution involves three key parameters: α (scale), β (shape), and γ (origin), which are estimated using the robust and straightforward L-moment procedure."*

4. One advantage of this dataset is high resolution. Therefore, the examples of spatial details could be shown and compared with other datasets.

Authors´ response:

- We also considered this a valuable suggestion, thanks to the reviewer's recommendation. As a result (**author's changes**), we have incorporated two high-resolution datasets into the evaluation for Central Asia and Africa (see the attached figures). Both the African and Central Asian datasets have been derived from CHIRPS data and exhibit a very high correlation when compared to MSWEP.

[Figure]

**Figure S2. Temporal correlation between the new high-resolution SPEI (SPEI-HR) based on (a) CHIRPS_GLEAM, (b) CHIRPS_hPET, (c) MSWEP-GLEAM, and (d) MSWEP-hPET, and Central Asia SPEI (CA-SPEI) for SPEI-1 month during the period 1981–2018.**

[Figure]

**Figure S3. Temporal correlation between the new high-resolution SPEI (SPEI-HR) based on (a) CHIRPS_GLEAM, (b) CHIRPS_hPET, (c) MSWEP-GLEAM, and (d) MSWEP-hPET, and Pan African SPEI (AF-SPEI) for SPEI-1 month during the period 1981–2016.**

5. The organization of this manuscript should be added to the end of the introduction.

Authors´ response:

- Thank you. Based on your recommendation we have added the organization of the manuscript at the end of the introduction as: (**author's changes**)
- *"The manuscript offers a comprehensive overview of the datasets and methodologies employed, outlined in Section 2. Section 3 provides an in-depth analysis of the results, coupled with a detailed discussion. The availability of the newly developed high-resolution datasets, along with relevant links, is covered in Section 4, while Section 5 delivers a concise conclusion."*

**Reference**

Peng, J., Dadson, S., Hirpa, F., Dyer, E., Lees, T., Miralles, D. G., Vicente-Serrano, S. M., and Funk, C.: A pan-African high-resolution drought index dataset, Earth System Science Data, 12, 753–769, https://doi.org/10.5194/essd-12-753-2020, 2020.

Pyarali, K., Peng, J., Disse, M., and Tuo, Y.: Development and application of high resolution SPEI drought dataset for Central Asia, Sci Data, 9, 172, https://doi.org/10.1038/s41597-022-01279-5, 2022.

We thank the reviewer for their time and valuable and detailed feedback. We have now addressed all comments and discussed them in the following.

The manuscript tries to develop five high-resolution (5 km) gridded drought records based on the Standardized Precipitation Evaporation Index (SPEI), which is essential and exciting for related fields. The structure of the paper is clear and the research questions are clear.

Authors´ response:

- Thank you for acknowledging the work and feedback

However, I think the introduction part needs to improve, and more details about the development of such a field need to be described.

Authors´ response:

- Thank you for the suggestion. We have added few lines on the need and development of the datasets, including the organization of the paper as recommended by reviewer#1.

The result and discussion should be separated and focus on the main result of the study.

Authors´ response:

- Thank you for the suggestion. We believe that combining both aspects would be beneficial for this data description paper. We are happy to split the results and discussion if you still believe it is necessary.

Besides, the structure of the result needs to improve significantly, such as in sections 3.1 to 3.3. A significant polish is needed.

Authors´ response:

- We have modified this section by adding more results, including additional evaluations with high-resolution Africa-based SPEI and Central Asia SPEI datasets, both of which are available at 0.05° resolution from 1981-2016 and 1981-2018, respectively in section 3.4. In addition, we have added a section (3.5) on uncertainties, where we discussed the source of uncertainties in detail.

Others, such as line 86-87 is it correct?

Authors´ response:

- Thank you for carefully reading the paper. You are right there is an error in line 86-87. This is now modified as: (**author's changes**) "**The precipitation and evapotranspiration datasets used in this study are widely used and generally reliable, although there may be some regions where their reliability is limited.**"

**Reply to Referee #3**

We thank the reviewer for their time and valuable and detailed feedback. We have now addressed all comments and discussed them in the following.

The manuscript produced global high-resolution (0.05°) and long-term (1981-2022) SPEI datasets. It is suitable for publication after a minor correction.

Authors´ response:

- Thank you for recognizing our efforts and providing feedback.

Line 191: Replace Table 1 with Table 2.

Authors´ response:

- Thank you for the thorough review. : (**author's changes**) We have replaced Table 1 with Table 2.

Line 192: Please justify the selection of a log-logistic probability distribution among others. How you attained the model parameters?

Authors´ response:

- The choice of the log-logistic distribution was informed by previous research (Vicente-Serrano et al., 2010; Beguería et al., 2014), which demonstrated its superior performance in generating SPEI series with standardized properties (mean = 0, SD = 1) when compared to other probability distributions.
- The log-logistic distribution involves three key parameters: α (scale), β (shape), and γ (origin). These parameters are estimated using the robust and straightforward L-moment procedure. Further details on the parameter computation process can be found in Vicente-Serrano et al. (2010).
- In order to enhance the clarity of this section, we have incorporated the following text into the paper.

**Author's changes:**

- *"The choice of the log-logistic distribution for SPEI is based on previous research (Vicente-Serrano et al., 2010a; Beguería et al., 2014), which demonstrated its superior ability to generate SPEI series with standardized properties (mean = 0, SD = 1) compared to other probability distributions. The log-logistic distribution involves three key parameters: α (scale),*

*β (shape), and γ (origin), which are estimated using the robust and straightforward L-moment procedure."*

Remove the underlined sign from the values given in Table 2.

Authors´ response:

- Thank you. The underlines are now removed from the numbers.

Beguería, S., Vicente-Serrano, S. M., Reig, F., and Latorre, B.: Standardized precipitation evapotranspiration index (SPEI) revisited: parameter fitting, evapotranspiration models, tools, datasets and drought monitoring, International Journal of Climatology, 34, 3001–3023, https://doi.org/10.1002/joc.3887, 2014.

Vicente-Serrano, S. M., Beguería, S., and López-Moreno, J. I.: A Multiscalar Drought Index Sensitive to Global Warming: The Standardized Precipitation Evapotranspiration Index, Journal of Climate, 23, 1696–1718, https://doi.org/10.1175/2009JCLI2909.1, 2010.